# Prevalence and Work-Related Factors Associated with Lower Back Musculoskeletal Disorders in Female Shellfish Gatherers in Saubara, Bahia-Brazil

**DOI:** 10.3390/ijerph16050857

**Published:** 2019-03-08

**Authors:** Maria Carolina Barreto Moreira Couto, Ila Rocha Falcão, Juliana dos Santos Müller, Ivone Batista Alves, Wendel da Silva Viana, Verônica Maria Cadena Lima, Paulo Gilvane Lopes Pena, Courtney Georgette Woods, Rita Franco Rego

**Affiliations:** 1Post-Graduate Program in Health, Environment, and Work, School of Medicine, Federal University of Bahia, Salvador 40026-010, Brazil; falcao.ila@gmail.com (I.R.F.); vonyssa@bol.com.br (I.B.A.); wendel_hp20@hotmail.com (W.d.S.V.); pena@ufba.br (P.G.L.P.); ritarego@ufba.br (R.F.R.); 2Department of Technology in Health and Biology, Federal Institute of Education, Science and Technology of Bahia, Salvador 40301-015, Brazil; muller_juliana@yahoo.com.br; 3Department of Statistics, Institute of Mathematics and Statistics, Federal University of Bahia, Salvador 40170-115, Brazil; vmcadena@gmail.com; 4Department of Environmental Science and Engineering, University of North Carolina at Chapel Hill, Chapel Hill, NC 27599, USA; courtney.woods@unc.edu

**Keywords:** musculoskeletal disorders, lower back pain, female artisanal fisher, shellfish gatherers

## Abstract

Lower back musculoskeletal disorders (MSDs) are an important public health problem and the leading cause of disability worldwide, but with yet unknown prevalence among shellfish gatherers. To investigate the prevalence and work-related factors associated with lower back MSD in a population of female shellfish gatherers, an epidemiological cross-sectional study was carried out in Saubara, Bahia-Brazil, in 2013. The Brazilian version of the Job Content Questionnaire (JCQ) and the Nordic Musculoskeletal Questionnaire (NMQ), in addition to a questionnaire containing the physical demands adapted to the artisanal work, were applied to a random sample of 209 female shellfish gatherers. The prevalence of lower back MSD was 72.7%. Using multivariate logistic regression, the shellfish gatherers who had worked for more than 26 years in the activity showed a prevalence of 1.22 (95% CI: 1.04–1.44) times higher compared to those unexposed. Lower back MSD was 1.24 (95% CI: 1.08–1.42) times higher among those more exposed to work sitting with trunk flexion. Those performed manual handling and muscle force with the arms had a prevalence ratio of 1.18 (95% CI: 1.01–1.39). These results show the need for greater awareness of health and social welfare factors impacting workers in small-scale fisheries and will promote the elaboration of health care policies for this occupational class.

## 1. Introduction

Lower back musculoskeletal disorder (MSD) is considered a significant public health problem and one of the main causes of incapacity or absence from work in the world [1,2]. In addition, it is noteworthy for its long duration, incapacitating character and the resulting granting of sickness benefits [1,3].

Studies indicate that certain occupational activities are associated with the development of lower back pain [4,5,6,7]. Activities which require physical demands such as trunk flexion and rotation, load handling (pushing, pulling and lifting), and having to remain seated for extended periods are all considered risk activities for the development of lower back MSD [5,6,7,8,9]. In addition to physical demands, psychological demands within the work environment have also been widely studied [10,11]. However, not all occupational categories have been the subject of work-related MSD studies, and this includes the category of artisanal fishers, particularly women, in developing countries.

Fishermen are one of the largest and most traditional categories of workers in the world, representing approximately 38 million people in 2014, of which 90% are engaged in artisanal fishing [12]. Of this total, 84% are in Asia, 10% in Africa and 4% in Latin America and the Caribbean (LAC) [12]. Brazil ranks fourth in fish production in the Latin America and Caribbean region [12]. In 2011, for every 200 Brazilians, one was an artisanal fisherman [13]. However, knowledge concerning the health and working conditions of fishermen and their families, especially of fisherwomen, is still very scarce [14]. Communities that make a living from artisanal fishing are frequently among the poorest in the population [13]. It is estimated that 5.8% of fishermen in the world earn less than $1 a day [12]. This level of income characterizes these populations among the poorest in the world.

The latest official data on Brazilian artisanal fishing from 2011 reveals generalized negligence of this sector. In 2003, the production of fish from Todos os Santos Bay (TSB) represented 33.22% of the production of the state of Bahia, located in Northeastern Brazil [15]. Saubara, one of the municipalities of the TSB region, plays an important role in this economic sector, which is dependent on the artisanal fishing sector. In 2010, Brazilian fish production reached 19th in the world rankings [16] and approximately 45% of the annual production was generated from artisanal fishing [13], while in the Northeast that percentage was even greater, 75% [17]. Therefore, it represents an important historic, social and economic sector for the development of this region and the subsistence of several traditional populations [18,19].

Shellfish gathering is a type of artisanal fishing, characterized by gathering seafood by hand or with rudimentary equipment, or by digging into the sand or mangrove mud to collect mollusks and crustaceans. In Brazil this has been an activity traditionally performed mainly by women [18,19]. The shellfish gatherers perform all the steps of the production process, including the organization and production of the working tools, there is no division of labor, and it is done manually [18,20]. During the activity, they are exposed to several ergonomic risks, such as load handling and repetitive movements to dig the shellfish in the sand during the collection, or shucking (extracting the meat from the shell). In addition, they need to remain squatting or sitting in a bent overposition for hours [18,20]. 

Although women represent 47% of this total work force, which in developing countries is the equivalent of 56 million jobs [12], they are frequently more disadvantaged and vulnerable, as well as being politically, socially and economically marginalized in relation to men [14,19]. The work they exercise includes domestic services as well as activities directly related to fishing, which characterize a double working day [19]. 

This article emerged in the context of a research line entitled “Health, Environment and Sustainability of Artisanal Fishers.” Research related to this study has sought to understand aspects of the work process and work environment of shellfish gatherers [18,20,21]. Pena et al. [18] identified ergonomic and physical risks, such as sun exposure without proper protection, high muscle overload, excessive repetitive movements, as well as poor sanitation conditions as important occupational exposures for shellfish gatherers. Falcão et al. [20] and Muller et al. [21], studied, respectively, the prevalence of MSD in the neck or shoulder and health-related quality of life (HRQOL) in the same population of female shellfish gatherers in Saubara. Falcão et al. [20] found a prevalence of MSD above 70% in all segments studied. Muller et al. [21] revealed that HRQOL is substantially less in this group of workers in relation to the general population. Physical health was particularly affected, which can be related to their physically demanding work [21]. These findings provide support for actions and proposals for environmental and health policies within the Brazilian Universal Health System (BUHS), for example establishing a health care network specifically for artisanal fishermen.

Research on the industrial sector of fishing in Denmark [22], Galicia (Spain) [23], United States [24] and Taiwan [25] report a high prevalence of musculoskeletal complaints. Lipscomp et al. [20] observed that lower back symptoms were the most common cause of work impairment, potentially due to their exhausting workload [22,23,24,25,26]. In addition to MSD, psychosocial stress [24,25], skin melanoma, bronchitis, emphysema, lung cancer and infectious diseases [27] and unintentional injuries [26] have also been reported.

According to the International Labor Organization (ILO), the work in fishing is one of the most dangerous and stressful activities in the world [28]. In addition, the consequences are still barely visible in terms of occupational diseases recorded, especially in women [14,19], as described above. Thus, studies such as this may highlight the association between MSD and shellfish gathering activity, contributing to improving the labor rights’ legislation [29].

To determine the factors that may be involved with MSD in shellfish gatherers, it is necessary to verify the particularities of the work they conduct. Therefore, this study aims to investigate the prevalence and work-related factors associated with lower back MSD in a population of female artisanal fishermen/shellfish gatherers in Saubara, Bahia-Brazil.

## 2. Materials and Methods

### 2.1. Sample and Area

An epidemiological cross-sectional study was carried out with a sample of female artisanal fishermen in Saubara, Bahia-Brazil. The project was approved by the Ethics and Research Committee (number 356.261) of the Medical School of Federal University of Bahia, and the consent of the subjects was obtained. 

The target population of the current study lives in the municipality of Saubara, Bahia. This municipality is situated 94 km from Salvador, has an area of 163 km^2^ and is located within the Atlantic Rainforest biome [30]. Saubara is home to approximately 11,279 inhabitants, of whom 48.9% are men and 51.1% are women [31]. 

Only fishermen who are registered with the state, often through a local association, are entitled to the benefits currently granted by Brazilian government programs. For this reason, we predict that a majority of residents engaged in fishing or shellfish gathering as their occupation are likely registered. The 568 fishermen who are registered in the Saubara Association of Shellfish Gatherers (SASG) make up 11% of the economically active proportion of the population (EAP), thus making them an important contributor to the economic viability of the municipality [30,31]. In Saubara women comprise roughly 75% (426) of registered fishermen, while men make up the remaining 25% (142). 

The sampling was random, simple and without replacement, in which each individual was drawn using a random number chart. The registration list was used for the random selection. To calculate the sample, a prevalence of 50% was used with an error of 5%. Maximum prevalence was used for this calculation, since the disease burden in this population was not known. The total population (*N*) of 426 female shellfish gatherers resulted in a minimum sample of 203 shellfish gatherers. The margin of loss or refusal of 3 to 10% was calculated and resulted in a final sample of 209 women, 3% more than the minimum anticipated for losses. The authors collaborated with the SASG to recruit workers to participate in the study. Selected workers were invited by phone, letter or in person. 

The following inclusion criteria were defined for the research participants: female, 18 years of age or older and working as a shellfish gatherer for at least one year. Shellfish gatherers who were selected but that during the period of the research were not engaged in the activity due to health problems resulting from the fishing activity, were not excluded from the survey. However, no cases with these characteristics were present in this study. This strategy aimed at minimizing bias of the survival effect of the healthy worker [32].

### 2.2. Measures

Data were collected in the period between 10 April and 10 May 2013. A structured questionnaire that was previously validated [4] was adapted to the work of shellfish gatherers [20]. The following items were included in the questionnaire: identification; socio-demographic characteristics; job information; current and past occupational history; time worked in shellfish gathering; daily work hours; lifestyle habits, including smoking, alcohol consumption, use of medication, and physical activities; comorbidities; housework; musculoskeletal symptoms; and physical and psychosocial demands at work. 

The questionnaire survey data were collected by trained interviewers, which makes it possible to clarify doubts, avoiding misclassification [33]. Most data were self-reported, except for weight, height and abdominal circumference (AC), which were directly measured, according to Table 1.

Weight and height measurements were used to calculate the Body Mass Index (BMI), and AC which assesses the accumulation of fat in the abdominal region. In addition, a pilot study was previously applied to ten fisherwomen to check the questionnaire readability, to identify potential logistics difficulties and estimate the time necessary to interview each woman in this study.

Physical demands at work were adapted, by the authors, for each stage of the shellfish gathering activity through a questionnaire elaborated by Fernandes [4], observing validity and reliability criteria based on Stock et al. [33]. The physical demand questionnaire involves questions about posture (sitting, squatting, standing, walking, arms raised above shoulder height, trunk flexion, trunk rotation, repetitive, precise and fine hand movements), muscle strength (muscle strength with arms or hands), load handling (pushing, pulling and lifting the load) and physical pressure (physical pressure with the hands on the work tool) at work. 

Questionnaires applied through interviews to characterize physical demands at work have often been used by epidemiologists [4,33]. The information obtained through the self-report of the worker, in scale form, with the duration, frequency and intensity of exposure have shown acceptable agreement when compared to observational studies or direct measures [1].

The psychosocial aspects were measured by means of scores obtained for psychological demands, control and social support at work, and collected through the Job Content Questionnaire—JCQ, with all questions checked for validity and reliability [34], as well as its Brazilian version [35]. According to Karasek [34], psychosocial demand refers to the demands of the task related to rhythm, concentration and time pressure; ability and creativity in the execution of tasks, and autonomy in the decisions of how to perform the work itself are reflexes of the control; and, social support takes into account support at work, either by the boss or by colleagues.

Information about musculoskeletal symptoms was collected through the extended version of the Nordic Musculoskeletal Questionnaire (NMQ) [36,37], with the inclusion of questions related to severity, duration and frequency of the symptoms [36]. The NMQ is an instrument used worldwide to assess MSD and was cross-culturally validated for the Brazilian population [38]. Results of Pinheiro et al. [33] survey revealed an 86% agreement between symptoms reported in the NMQ and the given respondent’s clinical history. More information can be found in Falcão et al. [16].

### 2.3. Definition of Lower Back MSD

The presence of pain or discomfort in the lower back in the past year was assessed along with the severity, duration and frequency of these symptoms [36,37]. Cases of lower back MSD were established for workers who reported pain or discomfort in the lower back over the last 12 months at work, which lasted a minimum of one week or at least a monthly frequency and was not caused by an acute injury [36]. The symptoms were associated with at least one of the following items: symptom severity rating ≥3, on a scale of 0 to 5 (no discomfort to unbearable pain); seeking medical attention for the problem; absence from work; change of work due to health restriction. Cases of lower back pain or discomfort referred to the complaint of pain in this region in the last 12 months, without the severity criteria described above.

To increase the specificity of the instrument, questions related to the severity, duration and frequency of the symptoms were included. We also chose to record pain in the last 12 months, for purposes of comparison with literature data, considering that some studies do not use severity criteria [4].

Lower back MSD corresponds to the dependent (outcome) variable in the current study. Variables that follow were considered as independent: sociodemographic; occupational; life habits; BMI; AC; children less than two years-old; physical demands; and psychosocial demands.

### 2.4. Exposure Definition

Physical demands at work were evaluated through questions answered by the workers related to the duration, frequency or intensity, on a scale ranging from 0 to 5. These answers represent the degree of evaluation that the female shellfish gatherer reported about her exposure [4]. 

Exposure to psychosocial demands was classified according to Devereux [39] as: 1. High exposure to psychosocial demands: high psychological demands, low control over work, and low social support; 2. Low exposure to psychosocial demands: low psychological demands, high control over work and high social support. At least two of these criteria must be met in both classifications for the shellfish gatherer to be rated in each group.

In Table 1, the independent variables, categories and classification criteria are presented. The variables were better detailed and had a stronger association when dichotomized in terms of the percentiles. The continuous variables, such as physical demands, years working, daily work hours, weekly housework and age were categorized as a function of the 25th, 50th and 75th percentiles.

### 2.5. Statistical Analysis

Statistical treatment was performed on the data collected from the questionnaire using the program R, the 2.15.2 version (Free Software Foundation, Boston, MA, USA). In the descriptive analysis of the data, the measures of central tendency (means, medians), dispersion (standard deviations), percentiles for the continuous variables and frequencies for the categorical variables, were calculated. The prevalence of lower back pain or discomfort over the last 12 months and the specific cases of lower back MSD were calculated.

The physical demands of posture, muscle strength, load handling and physical pressure were considered for the steps of collecting, transporting and shucking of the shellfish gathering. These stages were considered the most important, since they require more time to dedicate to the task and workload. Due to the great variability of shellfish activity, to summarize and reduce this set of physical demands variables, factor analysis of collection, transport and shucking stages was performed [44]. The physical demands in each stage of the shellfish gathering process that presented a Pearson correlation above 0.20 were selected for factor analysis. The definition for the number of factors was based on an eigenvalue ≥1 through the method for estimation of factorial loads. Varimax with Kaiser normalization was used as a rotation method [45]. 

To calculate the prevalence ratio and the confidence intervals, the risk factors of the continuous variables were categorized according to one of the percentiles 25, 50 and 75, with the aim of identifying what percentile better related to the response variable. In the bivariate analysis, the prevalence ratios and the confidence intervals (CI) of 95% were calculated.

The pre-selection of independent variables to enter the initial multiple logistical regression model was based on univariate logistical regressions, considering a *p*-value of less than 0.25 for the Wald test for coefficient significance [46]. The biological plausibility [1] of the associations was also considered.

The final model was obtained through the backward selection method, based on the likelihood ratio test and the Wald statistical test considering a significance level of 5%. Then, the delta method was used to calculate the adjusted prevalence ratios and their respective confidence intervals of 95% [47]. This method provided a good approximation for the averages, variances and co-variances of non-linear functions for one or more variables. Thus, for cross-sectional studies, a comparison of the results from the regression analysis and tabular analysis without using the Odds Ratio (OR) can be done, since they overestimate the punctual estimates as well as increase the inaccuracy of the confidence intervals [48].

No significant terms at the level of 5% were found during the analysis performed to identify confounding and interaction. For the final model, the Le Cessie and Houwelingen Test of Goodness of Fit [49] was performed, which showed a good fit for the model. The residual graphs constructed did not presents any observational discrepancy.

## 3. Results

The mean age was 39.6 years (±11.5 years). Approximately 96.0% of the sample declared self-identified their race as black or mixed-race (referred to as brown), and 74.6% have incomplete secondary education or less. The average income obtained by the female artisanal fishermen was equivalent to U.S. $ 67.69/month (Table 2).

The values found for prevalence of lower back pain or discomfort in the last 12 months and cases of lower back MSD were 82.8% (173) and 72.7% (152), respectively. A prevalence of 59.8% of pain in the last seven days was observed.

Table 3 reports the physical demands grouped in factors from a component matrix, corresponding to the stage of collection, transport and shucking. Collection was summarized by four factors which explained approximately 58.0% of the data variability. Factor 1 was load handling (pull, push and/or lift) described as physical pressure with hands on the work tool and use of arm-muscle strength; Factor 2 refers to the postures of squatting, trunk flexion and trunk rotation; Factor 3 refers to standing and walking; and Factor 4 represents repetitive hand movements and precise and fine movements. 

Transport was summarized by three factors, which together explained approximately 65.2% of the total data variability. Factor 1 was lifting the load, arms above shoulder height and use of arm-muscle strength. Factor 2 represents the physical pressure of the hands on the work tool and the pulling the load (pulling). Factor 3, the postures of walking and standing were most prominent. 

Factor analysis for shucking was also summarized in three factors, which explained approximately 65.5% data variability. Factor 1 incorporated the demands of trunk rotation, load handling (push, pull and lift), and physical pressure with hands on work tool and arm-muscle strength. In Factor 2, repetitive hand movements and precise and fine hand movements were prominent. Lastly, Factor 3 was sitting with trunk flexion (towards the ground).

The values for prevalence ratio (PR) and the confidence intervals of 95% (95% CI) are presented in Table 4. The data reveal a positive association between lower back MSD and age; years working; and sitting with trunk flexion (Factor 3 of shucking). Although physical demand related to Factor 1 of collection (Muscle strength, physical pressure with hands and load handling) was not statistically significant in the final model, it was shown to be a borderline variable at 95% CI (1.00–1.37). 

In Table 5 the final model obtained from the multivariate analysis is described. Lower back MSD was 1.18 (95% CI: 1.01–1.39) times more frequent among workers exposed to load handling, muscle strength with arms during collection of the shellfish, than of those not exposed. The shellfish gatherers who worked seated with the trunk flexion during the shucking stage had a lower back MSD prevalence 1.24 (95% CI: 1.08–1.42) times greater than those not exposed. Those that have been working in shellfish gathering for more than 26 years had a frequency for lower back MSD 1.22 (95% CI: 1.04–1.44) times greater than those who worked 26 years or less.

## 4. Discussion

The results presented in this study reflect a high prevalence of lower back pain in the past year, as well as for cases of lower back MSD in female shellfish gatherers of Saubara. This is higher than the prevalence of low back pain reported on other occupational groups in the literature, which varies from 24% to 59.4% [5,6,7,50,51], despite using a more rigid definition, revealing that this population is more exposed to risk factors related to the onset of back pain. These findings reveal the magnitude of this problem in fishery workers, particularly female shellfish gatherers, and is consistent with high physical workload reported in the literature [14,18,20,26]. A high prevalence of pain in the last seven days was observed, indicating that many shellfish gatherers work with the presence of symptoms. The main predictor of this disorder is related to their work, such as time working as shellfish gatherer, load handling and duration of sitting and flexed trunk posture. 

Surveys using the NMQ in the categories of male fishers [22] and commercial fishers [24] indicated 80% [22] and 52% [24] of respondents claiming lower back pain during the past year. Using other instrument, lumbar complaints in the past year was also assessed in female shellfish gatherers from Spain [23] and industry fishing workers from India [25], which identified a 65.5% [23] and 33% [25] prevalence, respectively. The discrepancy identified in the population of female industry fishing workers [25], compared to our results, is possibly related to the mean age (23 ± 6.4) and years working (2.8 ± 12.9) [25], which is significantly lower than that found in the current study sample. 

According to our results, those participants with 26 years or more experience in shellfish gathering suffered a higher prevalence of lower back MSD. The variable age (≥38 years), was statistically associated with this pathology in univariate analysis, however, it did not remain in final model. Although not statistically significant in the current survey, age is reported as an important factor for the development of lower back MSD, since the load capacity of the spine decreases over the years [1]. Rodriguez-Romero et al. [23] observed positive correlation between disability caused by lumbar pain and age and years worked. The increased risk to low back symptoms and age was also reported in rural fishing settlement [52] and fish processing [25]. These finding shows that the time of exposure to the activity of shellfish gathering contributes to lower back MSD, probably due to the trauma accumulated over the years [1]. 

Some factors may explain the high prevalence of lower back MSD identified in this study. Since this is a low-income population, they need to work for long periods to increase production and ensure survival of their family by selling or consuming the shellfish [14,18], even in the presence of fatigue and pain. Also, here are high physical demands required in the shellfish gathering activity [18,20,23], similar to other studies carried out among other workers in the fishing sector [22,24,25]. This exposure begins in childhood, in average 13 years (±7.2), with a minimum of 4 years [20]. In addition, women reported that they dedicate many hours per week to housework, characterizing a double work shift. These factors aggravate the physical burden on these women.

The results of this study indicate that a sitting posture and trunk flexion are associated factors for lower back MSD. No further survey in the fisherman category investigated the relationship of sitting in a static posture with MSD, however, other studies also suggest this positive association [53,54]. Previous studies carried out among other job sectors have reported that working in an abnormal posture (bending/twisting) increased the risk of developing lower back pain [4,50,51]. Prolonged periods with trunk flexion and rotation can provoke compression and protrusion of invertebrate discs of lumbar spine [55]. In addition, being seated with trunk flexion promotes maximum disc pressure and is described as one of the most impactful for lower back pain. 

This study also revealed that load handling and muscle strength with arms was associated with lower back MSD. This finding is supported by other studies which report that lifting, pulling and pushing objects are factors associated to lower back pain [5,6,7,8]. In addition, positive association between lower back pain and workload, such as material handling and non-neutral trunk posture (flexion, lateral bending and rotation) was found in fishermen from United States [51], Denmark [22] and Thailand [56] and women performing fish processing from India [25]. These findings bring forward the physically challenging work environment of fishing activities, whether industrial, commercial or artisanal.

In relation to psychosocial demands, they have also been widely studied in research about MSD. They include accelerated work rhythm, low social support, monotony, low control and high stress at work [1], high psychological demands, and low social support [39]. In the current study, psychosocial demands and lower back MSD were not significantly associated. The population of female shellfish is characterized by informal working hours and therefore offers high control over the exercise of their activities. In addition, most of the shellfish gatherers go to the mangrove in groups, which are chosen according to affinity [18], and, therefore, show a high degree of social support. However, research about the psychosocial demands in traditional populations is still scarce. Nag et al. [25], in a study with fish processing workers, concluded that 22.8% of the reported symptoms were explained by psychosocial variables. Though, they performed the study in a population of industrial workers [25], who suffer the same pressures to increase production as any formal worker, in a different manner than the population of the current study.

### 4.1. Strengths and Limitations

There are strengths in this study that can be cited, such as the successful application of the NMQ in a population with a low level of schooling. Another strength is that studies on occupational health of shellfish gatherers are rare, and the results of the present research can bring to light this important public health problem in a marginalized population, such as shellfish gatherer workers. 

Some reasons are cited for the occurrence of inconclusive associations. It may be a result of a small sample size, lack of variability of exposure, presence of another risk factor and/or failure to control confounders and categorization of data [45]. The lack of contrast required for exposure to risk factors makes it difficult to associate lower back pain with physical demands in very homogeneous populations. The limitation in establishing causality is inherent to the applied epidemiological method. However, these limitations should not be a barrier to interpretation of data from many epidemiological studies that identify associations, but rather as one of the limitations that exists in many areas of health-related research.

There are methodological differences in this research field, which limit comparing low back MSD studies and, consequently, the ability to assess the extent of the problem. The case definitions and diagnostic criteria are different between studies, even among those who also used the NMQ to assess MSD [18,20,21,22]. 

### 4.2. Recommendations 

Female artisanal fishermen are exposed to many varied occupational risks without appropriate protection of their health. Although representing almost 0.5% of the Brazilian population [10], very little action has been taken by the public Health Universal System (SUS) in Brazil to better protect those working in this job sector [24]. To reduce ergonomic exposures occurring among these traditional workers, the implementation of intersectorial actions for prevention, treatment and rehabilitation of MSD in primary health care is imperative. Simple interventions, such as guidelines for ways to handle loads in order to reduce stress to the lower back and provide more ergonomic benches with backrests can be of great value.

However, in order to reduce inequities in the health sphere of these workers, it is essential that these actions consider the specificities of the fishers, their artisanal work process and status as self-employed. It is important that authorities support associations of shellfish gatherers, to promote measures that improve the market value of their product and, consequently, increase the income of women working in this sector. This increase may reduce the need for long daily hours of work, as well as prevent these women from taking their children to help at work to increase productivity, and instead, take them to school, breaking the cycle of low income and low education. Therefore, there is a need for an intersectoral effort to establish more sustainable models of development that ensure the survival of shellfish gatherers and their families in technical, economic, social, educational, cultural and environmental conditions. The expectation is to give greater visibility to the challenges that artisanal fishermen face and to promote the development of public health policies targeted to them specifically.

Prospective epidemiological studies with artisanal fishermen are needed to confirm the cause-effect relationship between physical demands and MSD. Validation and reproducibility studies of the NMQ in this class of workers are of paramount importance in verifying the applicability of the instrument in a population with a low level of schooling. Research to evaluate the inability to work and its implications on quality of life are also important to characterize the occupational health scenario of this population and to contribute to the theoretical basis of the technical epidemiological link of MSD with shellfish gathering. 

## 5. Conclusions

As a result of this study, an important association between lower back MSD and work-related factors in shellfish gatherers was evident. Physical demands were positively associated with lower back MSD, and the most frequently associated demands were: load handling and use of arm-muscle strength during the collection of shellfish with rudimentary and improvised tools; and sitting with trunk flexion. Also, those who had spent more than 26 years as a shellfish gatherer presented higher rates of lower back MSD. In this study, psychosocial demands, non-occupational and individual factors were not associated with lower back MSD. Even with the adoption of severity criteria for the definition of cases of lower back MSD, a high prevalence among the shellfish gatherers in Saubara, Bahia, was found, which suggests that workers continue to perform their activities, even in the presence of pain. This recognition is a way to sensitize professionals of the public health system and experts from the social security department about the precarious work conditions and their impact on the health of shellfish gatherers, in order to promote better public policies for this population.

## Figures and Tables

**Table 1 ijerph-16-00857-t001:** Independent variables, categories and classification criteria.

Variable	Criteria Used	Risk	Non Risk
**Marital status (PBA *)**	Lives alone/with someone	Married, lives with partner.	Single, separated
**Schooling (PBA)**	Schooling level	<High School	≥High School
**BMI [40]**	Weight and height measured by anthropometric scale (kg/m^2^).	BMI ≥ 25	BMI < 25
**AC [41,42]**	Average point between the last costal arch and the upper iliac crest.	AC > 80 cm	AC ≤ 80 cm
**Alcohol [4]**	Frequency of consumption	≥1 time/week	<1 time/week
**Smoking [3]**	Smoker or non-smoker	Yes	No
**Current work [4]**	Have other job	Yes	No
**Previous work [4]**	Worked in another field.	Yes	No
**Daily work (PBA)**	Dichotomized through the median (p50)	>9 h	≤9 h
**Weekly housework (PBA)**	Dichotomized through percentile 75 (p75)	>24.75 h	≤24.75 h
**Years working (PBA)**	Dichotomized through p50.	>26 years	≤26 years
**Physical activity [23,43]**	1. ≥(3 times a week; 30 min) 2. <(3 times a week; 30 min) 3. Does not do	Options 2 or 3	Option 1
**Age (PBA)**	Dichotomized through p50	>38 years-old	≤38 years-old
**Children (PBA)**	Children ≤ 2 two years old	Yes	No
**Physical demands [44]**	A factor analysis was performed with the most significant variables for each stage of the process (collection, transport and shucking).		
**Collection (PBA)**	Factor 1: Load Handling; physical pressure with hands on tool; muscle strength with arms.	>p50 (−0.07)	≤p50 ( −0.07)
	Factor 2: Squatting; trunk flexion and trunk rotation.	>p50 (0.13)	≤p50 (0.13)
	Factor 3: Standing; walking	>p50 (−0.07)	≤p50 (−0.07)
	Factor 4: Repetitive, precise and fine hand movements.	>p50 (0.11)	≤p50 (0.11)
**Transport (PBA)**	Factor 1: Load lifting; arms above shoulders height; muscle strength with arms.	>p75 (0.75)	≤p75 (0.75)
	Factor 2: Physical pressure with hands on tool; pulling load.	>p75 (−0.76)	≤p75 (−0.76)
	Factor 3: Standing, walking.	>p75 (−0.76)	≤p75 (−0.76)
**Shucking (PBA)**	Factor 1: Trunk flexion; load handling, physical pressure with hands on tool; muscle strength with arms.	>p25 (−0.77)	≤p25 (−0.77)
	Factor 2: Repetitive, precise and fine hand movements.	>p75 (0.73)	≤p75 (0.73)
	Factor 3: Seated, trunk flexion.	>p50 (−0.23)	≤p50 (−0.23)
**Psychosocial demands [39]**	Dichotomized through the median calculation of each criterion. The fulfilling of at least two criteria characterizes the high demand.		
	Psychological demand	>34	≤34
	Control	≤66	>66
	Social support	≤13	>13

* Performed by the authors (PBA).

**Table 2 ijerph-16-00857-t002:** Socio-demographic characteristics of shellfish gatherers sample (*N* = 209) from Saubara, Bahia-Brazil.

Variables	Total	
	Mean	±DP
Age (years)	39.6	11.5
Monthly income (U.S. $)	67.69	51.69
**Variables**	***N***	**%**
**Race**		
Black	125	59.8
Brown	76	36.4
White	8	3.8
**Schooling**		
Did not study/primary education	48	23
Elementary school complete/incomplete	94	45
Incomplete secondary education	14	6.7
Secondary education	53	25.4

**Table 3 ijerph-16-00857-t003:** Results of factor analysis for the variables related to posture, muscular force and handling of the load.

Collection Stage	1	2	3	4	Transport Stage	1	2	3	Shucking Stage	1	2	3
Standing	−0.039	−0.006	**0.883**	−0.090	Standing	−0.112	0.311	**0.838**	Seated	−0.147	0.206	**0.332**
Walking	0.077	0.038	**0.735**	0.033	Walking	0.385	−0.276	**0.717**	Trunk flexion	0.599	−0.314	**0.714**
Squatting	0.026	**0.677**	−0.052	−0.230	Arms above shoulders height	**0.741**	−0.140	0.081	Trunk rotation	**0.671**	−0.081	0.036
Trunk flexion	−0.005	**0.509**	0.006	0.084					Repetitive hand movements	−0.104	**0.444**	0.220
Trunk rotation	0.099	**0.652**	0.036	0.438	Muscle strength with arms	**0.694**	0.230	0.107				
Precise and fine hand movements	0.196	0.138	0.069	**0.894**					Precise and fine hand movements	0.222	**0.758**	0.106
					Physical pressure with hands	0.473	**0.646**	0.003				
Muscle strength with arms	**0.465**	0.014	−0.067	0.157					Muscle strength with arms	**0.735**	0.428	0.012
					**Load Handling**							
Physical pressure with hands	**0.571**	−0.051	−0.090	0.081	Pull	0.001	**0.832**	0.074	Physical pressure with hands	**0.724**	0.371	−0.093
					Lift	**0.723**	0.182	−0.006				
**Load Handling**									**Load Handling**			
Push	**0.494**	0.460	0.190	0.012					Push	**0.683**	−0.243	−0.346
Pull	**0.755**	0.431	0.018	−0.034					Pull	**0.705**	−0.321	−0.420
Lift	**0.710**	−0.086	0.239	0.042					Lift	**0.720**	0.070	−0.080

Source: Authors’ Calculations. The bold data signals the result of the factorial analysis and the physical demands selected to be part of each factor.

**Table 4 ijerph-16-00857-t004:** Univariate analysis of the prevalence ratio between lower back MSD, socio-demographic variables, lifestyle and occupational habits, psychosocial and physical demands (*N* = 209).

Variables	Risk		Non Risk		PR	95% CI
	*N*	Prevalence (%)	*N*	Prevalence (%)		
Age	83	82.2	101	69.0	1.19	1.02–1.40
Schooling	114	77.0	38	71.6	1.08	0.89–1.31
Marital status	99	75.5	53	75.7	1.00	0.85–1.18
Children	14	70.0	138	76.2	0.92	0.68–1.24
Smoking	7	70.0	145	75.9	0.92	0.61–1.40
Alcohol	32	74.0	120	76.0	0.98	0.81–1.19
Physical activity	96	75.0	55	77.5	0.97	0.82–1.14
Daily work	69	78.4	82	74.5	1.05	0.90–1.22
Weekly housework	40	81.6	111	75.5	1.06	0.84–1.20
Years working	86	84.3	66	66.7	1.26	1.08–1.49
AC	109	75.7	38	76.0	1.00	0.83–1.19
BMI	101	75.4	47	77.0	0.98	0.81–1.18
Current work	43	71.7	109	77.3	0.93	0.78–1.11
Previous work)	89	76.7	63	74.1	1.04	0.88–1.22
Psychological demand	54	73.9	98	76.6	0.97	0.82–1.41
**Physical Demands**	***N***	**Prevalence (%)**	***N***	**Prevalence (%)**	**PR**	**95% CI**
**Collection Stage**						
**Factor 1.** Muscle strength, physical pressure with hands and load handling	83	81.4	69	69.7	1.17	1.001–1.37
**Factor 2.** Squatting, trunk flexion and trunk rotation	97	78.9	55	70.5	1.12	0.94–1.33
**Factor 3.** Standing and walking	80	77.7	72	73.5	1.06	0.90–1.24
**Factor 4.** Repetitive hand movements and precise and fine hand movements	77	79.4	75	72.1	1.01	0.94–1.29
**Transport Stage**						
**Factor 1.** Arms above shoulder height, muscle strength and load lifting	42	84.0	110	72.8	1.15	0.99–1.35
**Factor 2**. Pressure and pulling load	119	77.3	33	70.2	1.10	0.90–1.35
**Factor 3.** Standing and walking	117	77.0	35	71.4	1.08	0.08–1.31
**Shucking Stage**						
**Factor 1.** Trunk rotation, physical pressure with hands, muscle strength with arms and load handling	118	78.1	34	68.0	1.10	0.94–1.29
**Factor 2.** Repetitive hand movements and precise and fine hand movements	41	73.2	111	76.6	0.90	0.74–1.10
**Factor 3.** Seated and trunk flexion	84	81.6	68	69.4	1.18	1.01–1.38

Source: Authors’ Calculations.

**Table 5 ijerph-16-00857-t005:** Multivariate analysis of the adjusted Prevalence Ratio for lower back MSD and variables of the final model in a sample (*N* = 209) of female shellfish gatherers in Saubara, Bahia-Brazil.

Variables	PR Adjusted	95% CI
Factor 1 of collection stage (Load handling, physical pressure with hands on tool and muscle strength with arms)	1.18	1.01–1.39
Factor 3 of shucking stage (Seated and trunk flexion)	1.24	1.08–1.42
Years working in shellfish gathering (>26 years)	1.22	1.04–1.44

Source: Authors’ Calculations.

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
