# Peer review of "Prevalence and Work-Related Factors Associated with Lower Back Musculoskeletal Disorders in Female Shellfish Gatherers in Saubara, Bahia-Brazil"

_ijerph, 2019, doi:10.3390/ijerph16050857_

Round 1

Reviewer 1 Report

This epidemiological study aims to investigate the prevalence of MSD symptoms among the shellfish gatherers and find its relationship with physical and psychosocial risk factors. 

I found tthat this article might be intresting for the readers as it studied a high-risk occupational group that few studies reported the MSD prevalence and the risk factors among the professional group. The paper is well-written, and the robust statistical analysis was performed. I propose that the authors provide extra information in the method section about the questionnaire. Although the references are provided, the brief information about the physical and psychosocial dimensions studies facilitates reading of the paper. Furthermore, some extra information in the introduction, method, and discussion section are provided, which are unnecessary and might be removed. 

I did not understand very well the logic behind the Chart 1. How did you dichotimize different factors ? How did you chose a specific cut of point? Please provide more explanation for Table 1 in the results section. It is difficult to undestand for the readers why did you do the factor analysis ? 

I put some comments directely in the pdf file. Please consier them for revising your manuscript. 

I would like to accept this paper for publication after these revisions. 

Kind regards

Author Response

Please find our responses attached.

Reviewer 2 Report

General Comments

Very well written. The authors are concise and to the point!

I do highlight how well the Statistical Analyses was conducted and described

A criticism is that the data is now getting old (2013). A more timely publication would have been better. I would wonder if the findings are still the same?

Specific Comments

¨Chart I¨ should be called Table 1 ?

Line 234: should residue be ¨residual¨?

Line 247: removed ¨the¨ before Factor 2

Line 261: capitalize the f in Factor

Line 274: ¨groups¨

Line 281: should maintaining be duration (i.e. static flexion) ?

Line 288: which is significantly lower…

Line 305: not sure what is the relationship between consuming shell fish and LBP

Line 325: activities

Line 327: not sure what ¨disc suffering¨ refers to?

Line 333: greater load

Line 342: fish processing workers 

Line 349: the present research

Line 404: not sure what gravity criteria is?

Author Response

Please find our responses attached.
